# Brief communication

# "Loss and Damage from a catastrophic landslide in Nepal"

Kees van der Geest[1], Markus Schindler[1]

[1]United Nations University Institute of Environment and Human Security, Bonn, 53113, Germany

5  Correspondence to: Kees van der Geest (geest@ehs.unu.edu)

**Acknowledgements.** The authors wish to thank the Asia-Pacific Network for Global Change Research (APN) for funding the fieldwork of this study. Acknowledgements also go to the Integrated Development Society (IDS) in Nepal for organizing fieldwork logistics. The field team consisted of Sunita Bhattarai, Keshav Paudel, Anjan Kumar Phoju, Ranjita Thapa, Sangita Pokhrel, Ram Krishna Kunwar, Bal Bahadur Gurung and Maria Kibta. This paper greatly benefitted from comments of three
10  reviewers: Karen Sudmeier-Rieux, Pandit Chhetri and Erin Roberts.

**Abstract.** This brief communication reports key findings of a recent research that studied the impacts of the 2014 Jure landslide in Sindhupalchok (Nepal) and the effectiveness of household preventive and coping measures. The people-centered methods reveal not just what is lost in the disaster, but also how and why. A key finding of the household survey is that households in
15  higher income groups incurred higher losses in monetary terms, simply because they had more to lose. By contrast, lower-income households lost more in relative terms: The value of their losses amounted to 14 times their annual earnings. Many lower-income households will never fully recover from this blow to their livelihoods and well-being. The findings have important implications for discussions on loss and damage valuation, compensation and relief.

## 1 Introduction

20  ### 1.1 What happened?

On 2 August 2014, a major landslide struck in a densely populated area 80 km northeast of Nepal's capital Kathmandu, in Sindhupalchok District. With a death toll of 156, it was one of the deadliest landslides in Nepal's history. The landslide had a length of 1.26 km and was 0.81 km wide at the bottom. It destroyed all land, houses, properties and infrastructure in its path and created a 55m-high dam in the Sunkoshi River. Behind the debris dam, a 3 km long lake inundated houses, farms and a
25  hydropower plant. The Araniko Highway, Nepal's only road connection to China, was severely damaged, leading to nation-wide economic impacts.

Our research tests a new toolbox for assessing loss and damage from climate-related stressors in vulnerable communities (for more info, see van der Geest & Zeb, 2015). The toolbox was developed to support empirical research, which is crucial for

enhancing understanding of one of the most controversial topics in the climate change negotiations: loss and damage. We attempt to answer which losses and damages the landslide caused to households in the area and how effective their preventive and coping measures were. Loss and damage is defined as "adverse effects of climate-related stressors that have not been or cannot be avoided through mitigation and adaptation efforts" (van der Geest & Warner, 2015).

5 **1.2 Climate Change Attribution**

To what extent can landslides, such as the one we investigated, be attributed to anthropogenic climate change? This is an important question in the context of international negotiations under the United Nations Framework Convention on Climate Change (Parker et al., 2015). However, it is also a very complex question to answer. Climate science that focuses on attribution of extreme events to climate change is relatively new, but progressing fast (James et al., 2014). On the one hand, landslides 10 are often triggered by extreme rainfall events (Dahal & Hasegawa, 2008), and more intense monsoon rainfall has been found to lead to more frequent landslides (Petley et al., 2007). The 2014 Jure landslide was preceded by two days of torrential rainfall (141 mm), which triggered the landslide. On the other hand, a causal relationship between more frequent extreme rainfall events and climate change in the Himalaya has yet to be established (Huggel et al., 2012). While climate change alters the conditions that underlie the region's weather, other factors that caused the Jure landslide were unsustainable land use, the 15 absence of effective water-channeling mechanisms, a weak geology and steep slopes. Thus, although anthropogenic causes may have increased the likelihood of a landslide event, anthropogenic climate change cannot be pinpointed as its definitive cause. Most survey respondents who had lived in the area for at least 20 years observed an increase in landslide frequency (92.6%) and intensity (97.3%) over this period.

**2 Methodology**

20 The people-centered approach of this study was primarily based on a household survey with quantitative and qualitative assessments by 234 respondents. Beyond the survey, expert interviews, focus group discussions and secondary sources provided additional information and were used to validate survey findings. The questionnaire had three parts. Part 1 starts with basic socio-demographic data and then continues with questions about respondents' livelihood activities, income, assets and food security. These questions feed into a Multi-Dimensional Vulnerability Index (MDVI). Part 2 assesses the landslide losses 25 and damages that respondents incurred and the effectiveness and costs of the preventive and coping measures they adopted. Part 3 inquired about respondents' perceptions of vulnerability and their recommendations for future actions that could be taken by organizations or the government to better protect people against landslide impacts.

# 3 Results

## 3.1 Household Profile

The findings presented in this article are based on the 234 questionnaire interviews. Households in the research area were found to be headed predominantly by males (81.5%). The vast majority of households (94.4%) have at least three sources of income, one of which was usually farming (98.7%). Land ownership amounts to a median of 3,200 m² per household. Approximately three out of every four households (76.8%) live below a poverty line of $1.25 per capita per day. The median income of the area is even lower, with a daily per capita income of $0.6. Nearly a third of respondents (28.2%) has never been to school.

## 3.2 Preventive Measures

Most of the respondent households took preventive measures against impacts of landslides and other extreme events (65%). Among these households, 41.6% attempted to diversify their livelihoods by engaging in different economic activities, and 37.6% placed physical barriers, mostly gabions, on the hillsides (Figure 1). For each preventive measure, respondents indicated how effective they thought it had been at minimizing landslide impacts. House adjustments (using stronger building materials or moving to a safer location) and pro-active migration were seen as most successful (see the dots in Figure 1). Placing physical barriers and land-use adjustments, on the other hand, were the least successful measures. We generally found that respondents had not expected a landslide of this scale, which limited the effectiveness of preventive measures that households adopted.

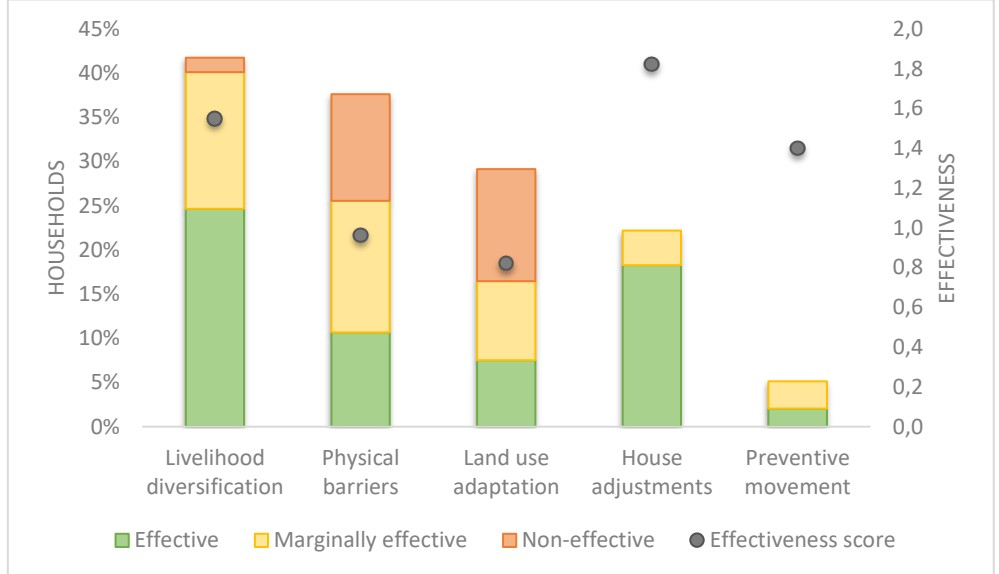

**Figure 1: Uptake and effectiveness[1] of preventive measures**

---

[1] *Effectiveness scores were calculated as 'effective'\*2 + 'marginally effective'\*1 + 'non-effective' \*0'.*

Preventive measures by organizations to minimize landslide impacts were rare. Only about a fifth of respondents stated that government agencies or non-governmental organizations took any preventive measures. Most of them mentioned that organizations had constructed gabions to prevent landslides. A few respondents also mentioned that organizations had planted trees to keep soil in place. After the landslide, an Early Warning System was established to warn settlements downstream

5 against a potential outburst flood from the debris dam.

### 3.3 Impacts

Likely due to the high prevalence of farming, the most common impact types were loss of crops (79.9%) and land (79.1%). Mental stress was reported by a majority of respondents (68.4%) and consisted of post-event trauma and fear of new landslides. In monetary terms, loss of land was the most severe impact type. For two thirds of the sample (66.6%), it exceeded $1000.

10

Households in the lowest income group were most severely affected by the landslide. The value of their losses amounted to 14 times their annual earnings (see Figure 2). Their potential for recovery is low: They may never return to the level of assets, livelihood security and well-being they had prior to the landslide. Households in the higher income group had higher absolute losses (median: $10,300), but the value of losses was much less in relative terms (three times the annual earnings). While

15 wealthier households may eventually recover from the impacts of the landslide, it will still take them years to restore their pre-landslide status.

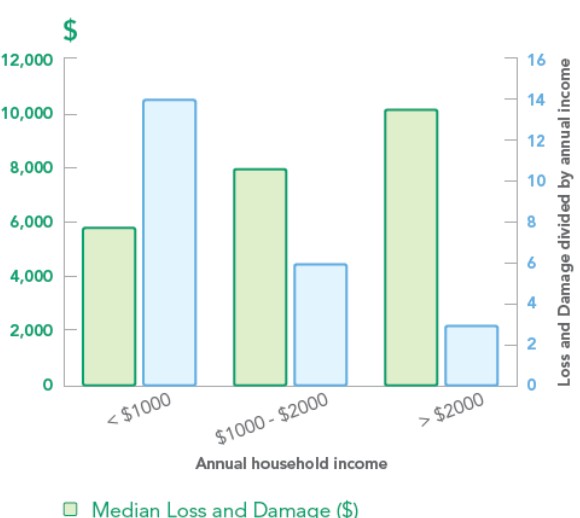

Figure 2: Monetary L&D by income group

As part of the questionnaire survey, respondents were asked about their perceptions of gender and age differences in the severity of landslide impacts. While a majority of respondents (59.8%) thought that men and women were equally affected, 29.1% believed that women were more affected. A common explanation was that men can run faster, and are more likely to escape when landslides hit. For differences in impacts between age groups, about half (51.3%) said that all groups were impacted equally. About a quarter (26.9%) thought that children suffered most from landslide impacts. This was mainly because on top of the other consequences, many children could not go to school for months after the landslide. Respondents who mentioned that elderly were affected most (10.7%) indicated that the elderly were not fast enough to escape and relocate to safer places. In sum, the majority thought that men and women and different age groups incurred similar landslide impacts but some respondents did identify differences.

## 3.4 Coping & Relief

More than three quarters of households adopted coping measures after the landslide (91.5%). Among these, households mostly received relief from organizations or the government (73.0%), survived on stored food or savings (63.2%) and engaged in migration (58.3%).

Selling assets and relying on social networks, loans, stored food and savings were the most effective coping measures (see figure 3). While some measures aided recovery, 54.5% said they will never recover from the impacts of the landslide.

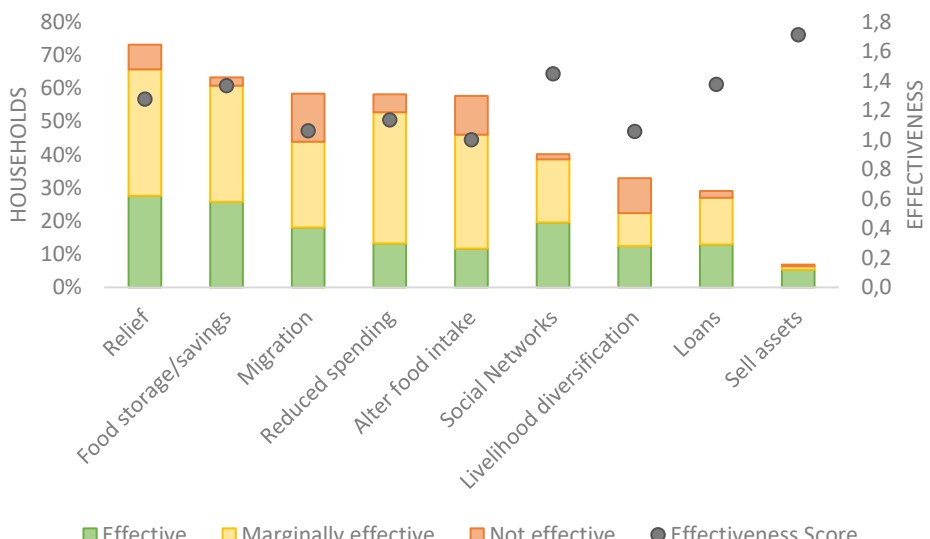

**Figure 3: Uptake and effectiveness2 of coping measures**

---

[2] *Effectiveness scores were calculated as 'effective'*2 + 'marginally effective'*1 + 'non-effective' *0'.*

Different organizations and the government provided relief to households, either in the form of monetary compensation (40,000 rupees per deceased household member) or in-kind aid. The government also commissioned construction work to reduce the risk of an outburst flood and remove debris. Respondents were generally positive and appreciative of these efforts, and recognized that it helped to mitigate landslide losses and damages. However, people also expressed concerns about a lack of transparency with regard to the distribution of aid. Particularly, respondents mentioned that well-connected households received more support than those who were in direst need.

## 4 Policy Recommendations

As part of the questionnaire, respondents were asked what the government and other organizations could do to minimize landslide impacts in the future. Respondents suggested that more gabions need to be placed on the hillsides to prevent landslides, people living in high-risk areas should be helped to resettle, awareness of landslide risk needs to be increased, and infrastructure needs to be improved to withstand landslides. Interestingly, respondents also argued for more scientific studies on landslides, which could help to reduce losses and damages in the future. To address losses and damages that could not be avoided, respondents called for adequate monetary compensation for lives and property lost.

Other possible direct improvements include more sustainable land use and planning, as well as better risk assessments of planned infrastructure projects. Further, landslide risks can be mapped through regular geological surveys and new scientific methods, such as the landslide susceptibility index (Shahabi & Hashim, 2015). In areas were adaptation is unlikely or hardly possible, the government could provide migration assistance to affected households. Indirect measures for enhancing coping capacity could include the provision of an insurance against potential damages from climate change in general and landslides in particular. Finally, promoting local households' diversification of income sources through micro-credit, education and vocational training would reduce people's vulnerability to natural hazards and increase their capacity to cope with impacts of idiosyncratic shocks.

## 5 Conclusion

This paper reported on research about impacts of the 2014 Jure landslide in Sindhupalchok District (Nepal) and the effectiveness of preventive and coping measures. The results indicate that attempts to prevent the landslide and minimize its impacts were suboptimal. At the same time, the difficulty in predicting where and when landslides will occur acts as a disincentive for households and organizations to commit scarce resources to prevention. Post-disaster relief, on the other hand, was heavily supported by organizations, and almost all households adopted coping measures to deal with landslide impacts.

Besides loss of life, houses and land, people in the area suffered a wide range of impacts from the landslide, particularly on their mental health and livelihoods. For global discussions on loss and damage valuation and compensation, the household impact analysis has an important conclusion: Expressing loss and damage in monetary terms to inform compensation mechanisms is likely to have an adverse outcome for the most vulnerable people. Households that are in direst need of support for survival and recovery would often end up receiving lower amounts of compensation than wealthier households whose absolute losses tend to be higher, simply because they have more to lose.

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
