# Peer review of "Brief communication"

_Natural Hazards and Earth System Sciences, 2016_

## Referee Comment (RC1) · K. Sudmeier-Rieux (Referee) · 18 Jul 2016

General comments The premise of this brief communication is highly praiseworthy: testing a new "tool kit" of loss and damage as applied to a case study, the Jure landslide in Nepal, one of Nepal's largest landslide disasters. The communication gives valuable information on losses occurred, coping strategies, methodological insights of the tool and policy recommendations. It is also written in clear and error free English.

A few overall comments: However the commentary as it reads lacks a number of key pieces of information in order to give the reader a clear understanding about the methodology, results and analysis. It is possible that the word limit of the brief commentary article format did not allow for more details, in which case authors should consider

submitting their research in another format as the article comes across as incomplete as currently published, with the conclusions not adequately substantiated.

Specific comments Section 1 Introduction It would be useful to mention why you developed a tool for loss and damage as this is one of the most controversial and discussed mechanisms of the international climate change agreements. This would also help to explain your entry point to the topic and why you have included section 1.2 Climate change attribution, which otherwise appears out of place. This section should be more balanced to include references to some of the work published by Petley (eg. Petley et al. 2007) which does attribute greater occurrence of landslides in Nepal to more intense monsoon rainfall. Before discarding the attribution to climate change, it would be useful to briefly summarize whether there was a rainfall event precedent the landslide and its intensity to understand the triggering mechanisms that led to the landslide.

2 Results As I am not so familiar with the "brief commentary" format of NHESS, I assume that the authors were not given the option of a "methodology" section or had no space to develop one? However it would have been useful to understand which households were selected and why. Other key questions: -how was the effectiveness scale established? - how is "successful" defined? Successful in reducing loss of lives, loss of property? Successful according to respondents or the researchers?

Another issue relates to the preventive measures that individual households undertook to reduce losses and many of the physical measures were considered "unsuccessful". What about the measures that government agencies undertook to reduce losses? Were there any and would these have been more successful?

3 Conclusions One piece of missing information relates to how much each household received in government compensation, which is usually standard, quite minimal and usually the same for each house, human life and livestock that was lost. So certainly it is true that poorer households may encounter more difficulties in recuperating but the statement that their losses will be compensated less as the value of their assets were

lower in monetary terms may not be accurate.

To sum, overall the contribution of this paper to advancing our understanding of various tools for assessing loss and damage is valuable but there are a number of gaps in the paper which need to be addressed despite the word limitations given by the brief commentary format.

———————————————————

---

## Referee Comment (RC2) · E. Roberts (Referee) · 29 Aug 2016

I very much enjoyed reading this paper. The findings are very interesting. However, I would like to have seen it go more in depth into the background, methods, findings and the possible policy implications if appropriate. In fact I felt there was scope for a longer paper. That said in its shorter version the paper could very much benefit from a bit more information in terms of the context (i.e. what are the climate trends in the region in terms of the timing and amount of annual rainfall), an overview of the methods and more space dedicated to findings (i.e. what were the implications of coping strategies? did response and relief strategies recognize and provide access to treatment for mental stress? what was the implication of households adopting erosive coping strategies).

[Figure]
Interactive
comment

It would also be useful to hear about the possible policy implications of these findings but that might be outside the scope of the paper. Some questions to consider are: what is the role of the state in the context of providing information about climate trends and supporting efforts to mitigate the risk of landslides? What policies could be put in place to prevent the poorest households from sliding further into poverty? However, this might be outside of the scope of a brief communication and/or space constraints might prevent adding too much more content to the paper.

Please also note the supplement to this comment:
http://www.nat-hazards-earth-syst-sci-discuss.net/nhess-2016-210/nhess-2016-210-RC2-supplement.pdf

**Supplement:**

[revised manuscript text omitted]

---

## Referee Comment (RC3) · R. Pandit Chhetri (Referee) · 4 Sep 2016

General Comments: This paper is well written and found it extremely interesting as the pragmatic application of loss and damage concept is limited (at least in Nepal). In a climate vulnerable country like Nepal where landslides are very common, this paper is a big resource. The paper has tried to capture the essence of loss and damage by developing a tool to assess it. The brief communication has looked into a specific disaster with aspects covering the demographic situations, the preventive measures and impacts with some coping mechanisms.

Given the weight of the paper and loss and damage concept, the length seems to be short leaving several questions unanswered due to limited explanation provided for the

readers. There surely is scope to add some additional information. For instance, the geography and climatic condition of Sindhupalchok (Nepal) could have been explained a bit.

The methodology of the paper seemed to have only covered the perception based study. Giving some science based input with climatic trends, geophysical study and landslide context in the hills of Nepal would have added more value to the paper. I acknowledge that the paper only being a brief communication could have resulted this. There is sufficient scope for the authors to capture these aspects in some form of longer publication.

Specific Comments: Like all good papers this has some limitations that I have identified and will mention the important ones below 1. Under 1.1, - Providing some general information about Sindhupalchok's geography and general landslide conditions of the district would help the readers understand the context better. For instance, is this the only landslide or is the district prone to such events historically (trend). The local respondents may have given the information. - The concept of loss and damage is mostly associated with the extreme events attributed to climate change so having some climate information of the district would be beneficial too. - There is the mention of a tool to asses the loss and damage but what does it contain? Can this be shown in a figure or box?

2. Under 1.2, - The paper mentions of landslides being attributed to climate change while giving some room for doubt. But, since the paper has also developed a tool to assess it, it would be good to give bit more explanations on the linkage between landslide and climate change, which would immensely help the readers to understand better especially from the affected areas and the authorities responding to it. - The information as it is under this paragraph seems to be bit isolated in the context of this paper if not linked to the problem (loss and damage) that the paper tries to address.

3. Under 2.1 and 2.2, - Can any reference be made to the level of climate change

knowledge of local people under the heading Household Profile? - I raise this because under 2.2 the paper mentions of people taking preventive measures to landslides and extreme events. Was this due to understanding of climate change impacts or just to respond to the potential risks that they face? - How frequent were the extreme events in the district?

4. Under 2.3 - Was there chance to disintegrate the impacts on women and children? Given such a massive disaster not only the poorest of the poor suffer heavily but there is equal chance that women and children were disproportionately affected too.

5. Under 2.4, - The on going coping contexts have been explained but did the respondents and local people have any alternative demand/solutions? Having this could add more value to the paper. - Could there be any suggestions provided by the authors as alternative measures and policy guidance for future?

---

## Author Comment (AC1) · 13 Sep 2016

Dear Dr. Sudmeier-Rieux,

Thank you very much for your constructive comments to our brief communication on Loss and Damage from a catastrophic landslide in Nepal. We have read your comments with great interest, and hope that our reply will be able to address them adequately.

As you have correctly suspected, we initially cut every corner of content on our submission in order to ensure that it complies with the formal limitations of the format. Fortunately, we have since learned that there is space to spare, so that we will be able

to ensure our readers' understanding of the methodology used, to go more into detail in the sections on climate change attribution, to mention measures by organizations and to address the issue compensation more thoroughly.

Your comment particularly highlighted to us that we need to be clearer in describing what we mean by the conclusion that those in direst need of support end up receiving the least because their losses are lower in monetary terms. As you correctly suspected, government compensation followed the general rule of being quite minimal and similar for each house, human life and livestock that was lost. However, our analysis highlights a clear discrepancy in the relative importance of quantifiable losses and damages, as higher losses in absolute terms sustained by wealthier households are likely to be less severe than the losses sustained by poorer households, albeit lower in absolute terms. Where our transcript lacked clarity is in emphasizing that the differences in compensation for loss and damage from natural disasters based on different needs and levels of income become particularly apparent from a global perspective. For example, a comparable disaster in the USA would lead to an estimation of monetary losses and damages much higher than in Nepal. Hence, what we are trying to do is to provide local evidence for a global discussion on the evaluation of and distribution of compensation for losses and damages from natural disasters.

Since we now have a better understanding of the format's formal limitations, we are confident that we will be able to fully incorporate your feedback into our manuscript, once the review-round is concluded.

Thank you again for your valuable inputs,

Kind regards, Markus Schindler & Kees van der Geest

---

## Author Comment (AC2) · 13 Sep 2016

Dear Ms Roberts,

Thank you very much for your valuable input.

In the brief communication, we refer to 'poor' households only in the abstract. In the actual body, we divide the sample into households in the 'lower' or 'higher' income group, based on the categories of annual income shown in figure 2. However, we recognize that these two different ways of describing households may be confusing, and will homogenize them for the final draft.

House adjustments are, very generally, modifications made to a house so that it is better equipped to withstand the impact of a potential landslide. This includes enhancing the physical prowess of the house, or moving it to a safer location. Land use adjustments, on the other hand, describe efforts to use the land in a way that avoids land degradation and the increase of landslide risk. Thank you for highlighting this, we will make sure to clarify the meaning of house adjustments in the revised text.

As you rightly point out, the landslide and its severity were caused by multiple factors. Heavy rainfall finally triggered the event to occur, but this would have been very difficult to predict before the landslide. Satellite images of the area show that signs of land degradation had already been appearing years before the landslide, but these signs tended to appear and disappear without any discernable pattern. Hence, predictability is a big issue that goes beyond our scope at the moment.

Finally, we would definitely like to include some policy recommendations. Initially, we assumed that the formal limitations of the brief communication format would prevent us from doing so, but we have since learned that we can expand the content a bit more once the review-round is concluded.

Thank you again for your comments and suggestions,

Kind regards, Markus Schindler & Kees van der Geest
* * *

---

## Author Comment (AC3) · 13 Sep 2016

Dear Mr Chhetri,

Your comments are highly appreciated, thank you very much.

You are correct in stating that there is sufficient scope for a longer publication. However, we intended to communicate our key findings as succinctly as possible. The brief communication format seems to suit this purpose perfectly. We nevertheless agree that some further explanation of Sindhupalchok's geographical/climatic history and situation, as well as the methodology, would be appropriate and will work to see this reflected in the revised text.

[Figure]

The toolbox to assess loss and damage contains information on how to plan, conduct and evaluate research on loss and damage. The content put forward in this toolbox area applied in the Nepal research, as a way to test its feasibility in the field. However, the toolbox is not yet publicly available, and we expect to publish it in late 2016. The overall knowledge on climate change by the respondents was limited, but people were very much aware of many of the environmental threats that applied specifically to them. Thus, their actions can be seen as a response to the potential risks that they face. As you correctly point out, we will clarify the linkage between landslides and climate change, and the connection of 1.2 to the overall L&D concept in the next version of the brief communication.

The data does not allow for an in-depth disintegration of impacts on women and children. However, we have information from open questions in our survey. These showed how the majority of respondents felt that while impacts were different for different age groups (e.g. no work for adults, no school for children), the impact was not necessarily 'worse' for a specific group.

Finally, respondents did offer input on how to improve on preventive and coping measures. If the scope allows, suggestions by the respondents, as well as us, will be included for policy guidance.

Thank you again for your valuable input,

Kind regards,

Markus Schindler & Kees van der Geest

---

## Author Response (AR1)

**Revision Report:**

**Brief communication**

**"Loss and Damage from a catastrophic landslide in Nepal"**

Kees van der Geest[1], Markus Schindler[1]

[1]United Nations University Institute of Environment and Human Security, Bonn, 53113, Germany

*Correspondence to*: Kees van der Geest (geest@ehs.unu.edu)

**Notes on the Revision Report:**
- Revised passages are written in *italics* and highlighted in grey
- Revisions in direct response to reviewer suggestions quote the corresopnding reviewer's feedback in a comment on the relevant section

[revised manuscript text omitted]

**Kommentiert [MS1]: Pandit Chhetri:**
"There is the mention of a tool to assess the loss and damage but what does it contain? Can this be shown in a figure or box?"

**Kommentiert [MS2]: K. Sudmeier-Rieux:**
"It would be useful to mention why you developed a tool for loss and damage as this is one of the most controversial and discussed mechanisms of the international climate change agreements"

**Kommentiert [MS3]: Pandit Chhetri:**
"The information as it is under this paragraph *[note: 1.2, Climate Change Attribution]* seems to be bit isolated in the context of this paper if not linked to the problem (loss and damage) that the paper tries to address. "

**Kommentiert [MS4]: K. Sudmeier-Rieux:**
"This section should be more balanced to include references to some of the work published by Petley (eg. Petley et al. 2007) which does attribute greater occurrence of landslides in Nepal to more intense monsoon rainfall."

**Kommentiert [MS5]: K. Sudmeier-Rieux:**
"Before discarding the attribution to climate change, it would be useful to briefly summarize whether there was a rainfall event precedent the landslide and its intensity to understand the triggering mechanisms that led to the landslide."

**Pandit Chhetri:**
"[...] it would be good to give bit more explanations on the linkage between landslide and climate change, [...]"

"The concept of loss and damage is          …

**Kommentiert [MS6]: Erin Roberts:**
"That said in its shorter version the paper could very much benefit from a bit more information in terms of the context (i.e. what are the climate trends in the region in terms of the timing and amount of annual rainfall)."

**Pandit Chhetri:**          …

**Kommentiert [MS7]: K. Sudmeier-Rieux:**
"However the commentary as it reads lacks a number of key pieces of information in order to give the reader a clear understanding about the methodology, results and analysis. "

*losses and damages that respondents incurred and the effectiveness and costs of the preventive and coping measures they adopted. Part 3 inquired about respondents' perceptions of vulnerability and their recommendations for future actions that could be taken by organizations or the government to better protect people against landslide impacts.*

**3 Results**

**3.1 Household Profile**

The findings presented in this article are based on the 234 questionnaire interviews. Households in the research area were found to be headed predominantly by males (81.5%). The vast majority of households (94.4%) have at least three sources of income, one of which was usually farming (98.7%). Land ownership amounted to a median of 3,200 m² per household. Approximately three out of every four households (76.8%) live below a poverty line of $1.25 per capita per day. The median income of the area is even lower, with a daily per capita income of $0.6. Nearly a third of respondents (28.2%) has never been to school.

**3.2 Preventive Measures**

Most of the respondent households took preventive measures against landslides and other extreme events (74.4%). Among these households, 41.6% attempted to diversify their livelihoods by engaging in different economic activities, and 37.6% placed physical barriers, mostly gabions, on the hillsides *(Figure 1)*. *For each preventive measure, respondents indicated how effective they thought it had been at minimizing landslide impacts.* House adjustments (enhancing the physical prowess of a house or moving it to a safer location) and pro-active migration were seen as most successful *(see the dots in Figure 1)*. Placing physical barriers and land-use adjustments, on the other hand, were the least successful measures. We generally found that respondents had not expected a landslide of this scale, which limited the effectiveness of preventive measures that households adopted.

**Kommentiert [MS8]: K. Sudmeier-Rieux:**
"how is "successful" defined? Successful in reducing loss of lives,
loss of property? Successful according to respondents or the researchers?"

[Figure]

**Figure 1:** Uptake and effectiveness[1] of preventive measures

*Preventive measures by organizations to minimize landslide impacts were rare. Only 21% of respondents stated that government agencies or non-governmental organizations took any preventive measures. Most of them mentioned that organizations had constructed gabions to prevent landslides. A few respondents also mentioned that organizations had planted trees to keep soil in place. After the landslide, an Early Warning System was established to warn settlements downstream against a potential outburst flood from the debris dam.*

**3.3 Impacts**

Likely due to the high prevalence of farming, the most common impact types were loss of crops (79.9%) and land (79.1%). Mental stress was reported by a majority of respondents (68.4%) and consisted of post-event trauma and fear of new landslides. In monetary terms, loss of land was the most severe impact type. For two thirds of the sample (67%), it exceeded $1000.

Households in the lowest income group were most severely affected by the landslide. The value of their losses amounted to 14 times their annual earnings (see Figure 2). Their potential for recovery is low: They may never return to the level of assets, livelihood security and well-being they had prior to the landslide. Households in the higher income group had higher absolute losses (median: $10,300), but the value of losses was much less in relative terms (three times the annual earnings). While wealthier households may eventually recover from the impacts of the landslide, it will still take them years to restore their pre-landslide status.
* * *
[1] *Effectiveness scores were calculated as 'effective'\*2 + 'marginally effective'\*1 + 'non-effective' \*0'.*

**Kommentiert [MS9]: K. Sudmeier-Rieux:** "how was the effectiveness scale established? "

**Kommentiert [MS10]: K. Sudmeier-Rieux:** "What about the measures that government agencies undertook to reduce losses? Were there any and would these have been more successful?"

[Figure]

$

**Figure 2: Monetary L&D by income group**

*As part of the questionnaire survey, respondents were asked about their perceptions of gender and age differences in the severity of landslide impacts. While a majority of respondents (59.8%) thought that men and women were equally affected, 29.1% believed that women were more affected. A common explanation was that men can run faster, and are more likely to escape when landslides hit. For differences in impacts between age groups, about half (51.3%) said that all groups were impacted equally. About a quarter (26.9%) thought that children suffered most from landslide impacts. This was mainly because on top of the other consequences, many children could not go to school for months after the landslide. Respondents who mentioned that elderly were affected most (10.7%) indicated that the elderly were not fast enough to escape and relocate to safer places. In sum, the majority thought that men and women and different age groups incurred similar landslide impacts but some respondents did identify differences.*

**3.4 Coping *& Relief**

More than three quarters of households adopted coping measures after the landslide (91.5%). Among these, households mostly received relief from organizations or the government (73.0%), survived on stored food or savings (63.2%) and engaged in migration (58.3%).

Selling assets and relying on social networks, loans, stored food and savings were the most effective coping measures (see figure 3). While some measures aided recovery, 54.5% said they will never recover from the impacts of the landslide.

**Kommentiert [MS11]: Pandit Chhetri:**
"Was there chance to disintegrate the impacts on women and children?
Given such a massive disaster not only the poorest of the poor suffer heavily but there
is equal chance that women and children were disproportionately affected too."

[Figure]

**Figure 3: Uptake and effectiveness2 of coping measures**

*Different organizations and the government provided relief to households, either in the form of monetary compensation (40,000 rupees per deceased household member) or in-kind aid to affected households in the area. The government also commissioned*
5 *construction work to reduce the risk of an outburst flood and remove debris. Respondents were generally positive and appreciative of these efforts, and recognized that it helped to mitigate landslide losses and damages. However, people also expressed concerns about a lack of transparency with regard to the distribution of aid. Particularly, respondents mentioned that well-connected households received more support than those who were in direst need.*

*4 Policy Recommendations*

10 *As part of the questionnaire, respondents were asked what the government and other organizations could do to minimize landslide impacts in the future. Respondents suggested that more gabions need to be placed on the hillsides to prevent landslides; people living in high-risk areas should be helped to resettle; awareness of landslide risk needs to be increased; and infrastructure needs to be improved to withstand landslides. Interestingly, respondents also argued for more scientific studies on landslides, which could help to reduce losses and damages in the future. To address losses and damages that could*
15 *not be avoided, respondents called for adequate monetary compensation for lives and property lost.*

*Other possible direct improvements include more sustainable land use and planning, as well as better risk assessments of planned infrastructure projects. Further, landslide risks can be mapped through regular geological surveys and new scientific*
* * *
2 *Effectiveness scores were calculated as 'effective'*2 + 'marginally effective'*1 + 'non-effective' *0'.*

**Kommentiert [MS12]: K. Sudmeier-Rieux:**
"One piece of missing information relates to how much each household
received in government compensation, which is usually standard, quite minimal and
usually the same for each house, human life and livestock that was lost."

**Kommentiert [MS13]: Erin Roberts:**
"It would also be useful to hear about the possible policy implications of these findings
but that might be outside the scope of the paper."

**Pandit Chhetri:**
"The on going coping contexts have been explained but did the respondents and local people have any alternative demand/solutions? Having this could add more value to the paper. - Could there be any suggestions provided by the authors as alternative measures and policy guidance for future?"

*methods, such as the landslide susceptibility index (Shahabi & Hashim, 2015). In areas were adaptation is unlikely or hardly possible, the government could provide migration assistance to affected households. Indirect measures for enhancing coping capacity could include the provision of an insurance against potential damages from climate change in general and landslides in particular. Finally, promoting local households' diversification of income sources through micro-credit, education and vocational training would reduce people's vulnerability to natural hazards and increase their capacity to cope with impacts of idiosyncratic shocks.*

**5 Conclusion**

*This paper reported on research about impacts of the 2014 Jure landslide in Sindhupalchok District (Nepal) and the effectiveness of preventive and coping measures.* The results indicate that attempts to prevent the landslide and minimize its impacts were suboptimal. At the same time, the difficulty in predicting where and when landslides will occur acts as a disincentive for households and organizations to commit scarce resources to prevention. Post-disaster relief, on the other hand, was heavily supported by organizations, *and almost all households adopted coping measures to deal with landslide impacts.*

Besides loss of life, houses and land, people in the area suffered a wide range of impacts from the landslide, particularly on their mental health and livelihoods. For global discussions on loss and damage valuation and compensation, the household impact analysis has an important conclusion: *Expressing loss and damage in monetary terms to inform compensation mechanisms is likely to have an adverse outcome for the most vulnerable people. Households* that are in direst need of support for survival and recovery would often end up receiving *lower amounts of compensation than wealthier households whose absolute losses tend to be higher, simply because they have more to lose.*